# Safety of Rotavirus Vaccination in Preterm Infants Admitted in Neonatal Intensive Care Units in Sicily, Italy: A Multicenter Observational Study

**DOI:** 10.3390/vaccines11040718

**Published:** 2023-03-23

**Authors:** Claudio Costantino, Arianna Conforto, Nicole Bonaccorso, Livia Cimino, Martina Sciortino, Mario Palermo, Kim Maiolo, Lucia Gabriella Tina, Pasqua Maria Betta, Mariacarmela Caracciolo, Carmine Mattia Loretta, Alessandro Arco, Eloisa Gitto, Salvino Marcello Vitaliti, Domenica Mancuso, Giuliana Vitaliti, Vincenzo Rosella, Giuseppa Pinello, Giovanni Corsello, Gregorio Serra, Bruna Gabriele, Fabio Tramuto, Vincenzo Restivo, Emanuele Amodio, Francesco Vitale

**Affiliations:** 1Department of Health Promotion Sciences, Maternal and Infant Care, Internal Medicine and Excellence Specialties (PROMISE) “G. D’Alessandro”, University of Palermo, 90127 Palermo, Italy; 2Regional Health Authority of Sicily, Via Vaccaro 5, 90145 Palermo, Italy; 3Neonatal Intensive Care Unit, Garibaldi Hospital, 95124 Catania, Italy; 4Neonatal Intensive Care Unit, University Hospital of Catania (G. Rodolico), 90123 Catania, Italy; 5Neonatal Intensive Care Unit, University Hospital of Messina, 98124 Messina, Italy; 6Neonatology Unit, NICU and Creche, ARNAS Civico, 90127 Palermo, Italy; 7Neonatal Intensive Care Unit, Maternal and Child Department, Buccheri La Ferla Fatebenefratelli Hospital, 90123 Palermo, Italy; 8Neonatology and Neonatal Intensive Care Unit, University Hospital of Palermo (P. Giaccone), 90127 Palermo, Italy

**Keywords:** rotavirus, rotavirus gastroenteritis, hospitalizations, rotavirus vaccination, preterm infants

## Abstract

Rotavirus (RV) is among the most common vaccine-preventable diseases in children under five years of age. Despite the severity of rotavirus pathology in early childhood, rotavirus vaccination for children admitted to the neonatal intensive care unit (NICU), who are often born preterm and with various previous illnesses, is not performed. This multicenter, 3-year project aims to evaluate the safety of RV vaccine administration within the six main neonatal intensive care units of the Sicilian Region to preterm infants. *Methods:* Monovalent live attenuated anti-RV vaccination (RV1) was administered from April 2018 to December 2019 to preterm infants with gestational age ≥ 28 weeks. Vaccine administrations were performed in both inpatient and outpatient hospital settings as a post-discharge follow-up (NICU setting) starting at 6 weeks of age according to the official immunization schedule. Any adverse events (expected, unexpected, and serious) were monitored from vaccine administration up to 14 days (first assessment) and 28 days (second assessment) after each of the two scheduled vaccine doses. *Results*: At the end of December 2019, 449 preterm infants were vaccinated with both doses of rotavirus vaccine within the six participating Sicilian NICUs. Mean gestational age in weeks was 33.1 (±3.8 SD) and the first dose of RV vaccine was administered at 55 days (±12.9 SD) on average. The mean weight at the first dose was 3388 (SD ± 903) grams. Only 0.6% and 0.2% of infants reported abdominal colic and fever above 38.5 °C in the 14 days after the first dose, respectively. Overall, 1.9% EAEs were observed at 14 days and 0.4% at 28 days after the first/second dose administration. *Conclusions*: Data obtained from this study confirm the safety of the monovalent rotavirus vaccine even in preterm infants with gestational age ≥ 28 weeks, presenting an opportunity to improve the vaccination offer both in Sicily and in Italy by protecting the most fragile infants who are more at risk of contracting severe rotavirus gastroenteritis and nosocomial RV infection.

## 1. Introduction

Worldwide, rotavirus (RV) is considered the main cause of severe gastroenteritis and mortality among children under five years of age [1,2,3,4]. Before the introduction of RV vaccines, RV was responsible for about 3.6 million cases annually of gastroenteritis among children 0–59 months, including 87,000 hospital admissions and around 700,000 medical consultations in Europe [5]. RV vaccination is strongly recommended by international health authorities and represents the most effective strategy for reducing rotavirus gastroenteritis (RVGE) among children [6,7].

In a literature review and meta-analysis by Burnett et al., it turned out that in studies of monovalent human-derived live attenuated RV vaccine (RV1), the median overall vaccine effectiveness against laboratory-confirmed RV diarrhoea in any age group was 83% (IQR 78–91) in low-mortality countries, 67% (40–75) in medium-mortality countries, and 58% (54–64) in high-mortality countries. Meanwhile, in studies of pentavalent human bovine reassortant RV vaccine (RV5), the median overall vaccine effectiveness was 85% (81–92) in low-mortality countries and 45% (44–57) in high-mortality countries. The median overall vaccine effectiveness in low-mortality countries was similar for RV1, RV5, mixed series (86%; 70–91), and non-product-specific (89%; 75–91) vaccinations [8,9].

Prior reviews have emphasized that the benefit of rotavirus vaccines is greater than the small risk of intussusception [10,11].

Several factors predispose premature infants to severe gastrointestinal illness. Delayed enteral feedings and high rates of histamine-2 blocker and proton pump inhibitor use reduce gastric acidity, a potent innate immune defense against rotaviral illness [12]. Once infected, premature infants exhibit more severe rotavirus infection than term infants [13,14,15]. Greater surface-to-volume body ratio increases the risk for dehydration and subsequent morbidity including seizures, and prematurity remains a risk factor for rotavirus-associated hospitalization [16,17].

Preterm infants are at particular risk of severe RV infection, with increased hospitalization rates, increased intestinal dilatation, abdominal distension, and mucoid stools [18,19]. Very low birth weight infants have a 2.6-fold increase in hospitalization compared to their term counterparts [20]. Protection of this group through vaccination should therefore be a high priority. Preterm infants exhibit reduced maternal antibody titers because placental transmission of immunoglobulin G (IgG) begins from 28 weeks gestation and the highest antibody transfer rates occur after 36 weeks. Longer hospital stays, multiple comorbidities, and low birth weight increase the risk of hospital-acquired infection.

Despite the severity of rotavirus pathology in early childhood, rotavirus vaccination for children hospitalized in neonatal intensive care units (NICUs), often born preterm and with several previous pathologies, is not usually performed [21].

The effectiveness of the RV vaccine in preterm infants has already been demonstrated by numerous studies. In an earlier study carried out by our working group, we observed that after the introduction of universal mass vaccination, there was a significant decline in hospitalizations coincident with an increase in vaccination coverage. In particular, the lowest number of RVGE hospitalizations was in 2016 (n = 156) when vaccination coverage was highest (45%) [22]. A study by Vesikari et al. demonstrated a reduction in hospitalizations and emergency department visits related to G1-G4 rotavirus gastroenteritis occurring 14 or more days after the third dose by 94.5% (95% CI, 91.2 to 96.6 percent) [7].

Many studies conducted in the US and Europe confirm that safety and tolerability of scheduled vaccines show no differences in preterm and full-term infants, and live attenuated vaccines are also considered safe [23,24,25,26].

Sicily is an Italian region where the Regional Health Department introduced the universal RV vaccination program (URV) into the immunization schedule in January 2013; it was the first region in Italy into which this program was introduced [10]. In 2017, RV universal mass vaccination (UMV) was introduced in Italy countrywide [27].

Sicilian RV vaccination effectiveness data show a significant reduction in the number of annual hospitalizations, hospitalization rate, and increase in median age at admission when comparing hospitalizations for RVGE in pre- (2009–2012) and post- (2013–2017) UMV introduction periods [22].

Since the implementation of vaccination starting in 2013, a progressive decrease in the number of hospital admissions for RVGE among children under five years of age has been observed in Sicily [22]. Unfortunately, vaccination coverage is still relatively low and below the official target of 95% (it was 59.5% in the 2019 birth cohort) [28].

The aim of this multicenter study was to provide additional data on the safety of RV vaccination in a critical population, such as preterm infants.

## 2. Materials and Methods

In Sicily, there are 47 public or private accredited clinics that perform deliveries.

An observational, descriptive, non-controlled, and non-randomized study was designed in order to evaluate the safety of neonatal intensive care unit administration of RV1 vaccination.

This study was conducted among the six larger NICUs in Sicily (three located in Palermo—PA, two in Catania—CT, and one in Messina—ME) from April 2018 to December 2019.

In this region, the Buccheri la Ferla Fatebenefratelli Hospital of Palermo is confirmed first for volume of deliveries (with 2255 deliveries carried out in 2017). Second in the ranking is the New Garibaldi Hospital—Nesima of Catania (2138 deliveries), followed by the Gaspare Rodolico Hospital Presidio of Catania (third with 1986 deliveries) and the Civico and Benfratelli Hospital Presidio of Palermo (fourth with 1804 deliveries). The University Hospitals of Palermo and Messina performed in 2017 612 and 1317 deliveries, respectively.

In Sicily, each minor local health unit (Agrigento, Caltanissetta, Enna, Ragusa, and Siracusa e Trapani) has its own NICU. The six NICUs described are the only public NICUs in the three Sicilian main LHUs (Catania, Messina, and Palermo) and collect most of the regional deliveries. There are no private NICUs; in Sicily, in fact, newborns from private clinics always cared for in the public NICUs.

The scientific coordination of the project was in the charge of the Department for Health Promotion, Mother to Child Care, Internal Medicine and Excellence Specialties of the University of Palermo and of the Regional Health Department of Sicily.

Due to the study design, no predetermined sample size was planned.

RV1 was selected for this study as it is the RV vaccine awarded by the regional tender for URV. It is a live and attenuated human strain that is to be administered orally in 2 doses at least 4 weeks apart starting at 6 weeks of age and to be completed by week 24 [21].

RV1 was administered to preterm newborns with gestational age ≥ 28 weeks from 6 weeks of age in accordance with the Summary of Product Characteristics (SmPC), after informed consent was obtained from parents. The protocol included the following exclusion criteria: unstable clinical conditions, such as cardiorespiratory support, parenteral nutrition, and antibiotic therapy for ongoing infection episode; previous intestinal invagination; necrotizing enterocolitis (NEC); intestinal malformations; and immunodeficiencies.

Vaccine administrations were performed in both inpatient and outpatient hospital settings as a post-discharge follow-up (NICU setting) [21].

Adverse events (expected, unexpected, and serious) were monitored from vaccine administration up to 14 days (first assessment) and 30 days (second assessment) after each of the two doses. In accordance with Directive 2001/20/EC Art. 2, adverse events were defined as:-*Expected adverse event*: expected reaction listed in the data sheet;-*Serious unexpected adverse event*: a serious adverse reaction whose nature, severity, or outcome is not consistent with the reference safety information;-*Serious adverse event*: any harmful clinical event that, regardless of the dose, requires hospitalization or prolongs ongoing hospitalization, results in severe or prolonged disability or incapacity, results in a congenital anomaly or birth defect, is life-threatening, or causes death [29].

Active surveillance was carried out by parents, who filled out a diary card that was returned at the time of the following outpatient visit or reported by telephone. If the child was hospitalized, active surveillance was carried out directly at the hospital by health care personnel. Parents were also asked to complete the diary card if adverse events required pediatric counseling.

The diary card was divided into sections within which detailed instructions on how to fill them out properly were included. All adverse events were to be indicated as absent, mild (does not require pediatrician consultation), moderate (requires pediatrician consultation), or severe (requires hospitalization), and parents were required to accurately indicate date of onset and end of symptoms.

Parents were informed and provided with informational materials on all adverse events in the data sheet, along with the most common symptoms for diarrhea (stools that are less formed and/or more watery than usual produced suddenly and/or with some daily repetition), urticaria (reddened (erythema), raised (pomphi), and itchy areas that may also affect mucous membranes (e.g., lips and eyelids)), and intestinal invagination (vomiting, emission of stool mixed with mucus and blood, abdominal pain, palpable mass in the abdomen, and drowsy state) in order to more easily recognize these adverse events and to alert their pediatrician.

The assessment of abdominal pain, irritability, and colic of the newborn was affected by the subjectivity of the parents and, therefore, was not objective.

Parents were also provided with a thermometer to identify possible systemic temperature elevations on a daily basis.

Based on data included in the SmPC, the following adverse events were monitored:

(1) Diarrhea, irritability (labelled as “common” in the SmPC);

(2) Abdominal pain, flatulence, skin inflammation (labelled as “uncommon” in the SmPC);

(3) Urticaria, intestinal invagination, blood in stool, and in very premature babies, longer than normal intervals between breaths (labelled as “very rare” in the SmPC);

This study was approved by the Ethical Committee of the University Hospital of Messina with resolution protocol number 526 of the 6 of April 2018. Moreover, the Sicilian Health Department and the Sicilian Regional Vaccination Technical Committee approved the extension of the project to all Sicilian hospitals involved in this study with resolution protocol number 15094 of the 22 of April 2018.

### Statistical Analysis

Absolute and relative frequencies were calculated for the categorical (qualitative) variables.

All data were entered into a database created with EpiInfo 3.5.4 (Centers for Disease Control and Prevention, Atlanta, GA, USA). All data were analyzed using the statistical software package Stata/MP 14.1 (StataCorp LP, College Station, TX, USA).

Adverse event rate was calculated as the overall number of adverse events occurring within 28 days after any dose, per number of doses administered.

## 3. Results

The observational period started in July 2018 and ended in December 2019. A total of 449 (of 485 potentially enrollable preterm newborns but for whom some parents did not provide consent; RR 92.6%) preterm newborns were enrolled in this study and fully vaccinated (two dose completion) with RV1. In Table 1, the main characteristics of preterm newborns enrolled at first administration of RV1 are reported. There were 223 (49.7%) male and 226 (50.3%) females, and the average weight of the newborns at first dose administered was 3388.5 g (SD: ±903.4). A large majority of enrolled newborns were both breastfed and artificially fed (212; 50.4%), while 35 (8.3%) were only breastfed and 174 (41.3%) were artificially fed. The main comorbidities observed at birth were moderate preterm birth in 241 (53.7%), very preterm birth (less than 32 weeks) in 105 (23.4%) of vaccinated newborns, and twinning in 47 (20.6%). In total, 36 (15.8%) of the preterm newborns were born with a respiratory distress syndrome (RDS), 10 had hypoglycaemia (4.4%), 5 had hypocalcaemia (2.2%), and 5 (2.2%) were small for gestational age (SGA).

Table 2 shows the characteristics of newborns enrolled per NICU. Mean gestational age in weeks was 33.1 (±3.8 SD) and the first dose of RV vaccine was administered at 55 days (±12.9 SD) on average. Overall, 8% of preterm newborns were first vaccinated in hospital wards/NICUs, and the remaining vaccinations were administered in outpatient visits during biweekly routine checks.

For the second dose, 62.8% of newborns received it in outpatient care, and the remaining 37.2% received it in territorial vaccination clinics of their respective local health unit.

As reported in Table 3, only 0.2% of vaccinated newborns reported fever >38.5 °C, and 0.6% and 1.1% had abdominal colic and diarrhoea, respectively, in the 14 days after first/second dose. Considering the 28 days after vaccination, only 0.4% of preterm newborns vaccinated had diarrhoea. Adverse events were self-limiting and resolved within 24 to 48 h after onset. The clinical conditions of the children who reported adverse events were, at the pediatric examination that occurred at the time of enrollment, comparable to those of the rest of the enrolled subjects.

Overall, no further expected/unexpected or serious adverse events were observed among the enrolled population.

The overall rate of adverse events was 161.8 per 100,000 doses administered.

## 4. Discussion

We evaluated the RV1 safety in an active surveillance system and the insurgence of adverse events during assessments 14 and 28 days after the administration of each of the two RV1 doses.

Of the 449 preterm infants enrolled in this study, only 1.9% experienced adverse events within 14 days after one of the two administrations (fever ≥ 38°: 0.2%, abdominal pain: 0.6%, and diarrhea: 1.5%). Our results are in line with the study from Goveia et al. where 2070 of the 68,038 children enrolled in the trial were between 25 and 36 weeks gestational age. Children were monitored 42 days after vaccination for intussusception and 7 days after vaccination for adverse events, such as fever, vomiting, abdominal pain, and behavioral changes. No statistically significant events were found between the vaccine and placebo groups [15,30].

The clinical relevance of RV infection in preterm infants and the availability of an effective vaccine has led health authorities to recommend RV immunization in all preterm infants regardless of gestational age as long as clinical conditions are stable [30,31]. Unfortunately, the first dose of the vaccine is rarely administered in a neonatal intensive care setting, so vaccination is delayed or even not carried out at all.

In our study, only 8.1% of the enrolled newborns started the rotavirus vaccination cycle within their NICU stay. Such a choice could have been determined by the uncertainty of RV horizontal transmission within the NICU ward. In order to increase vaccination coverage, it is important to increase neonatologists’ knowledge of the safety and efficacy of the RV vaccine [32].

Some authors have highlighted the importance of nosocomial RV (nRV) disease as an emerging public health issue [33]. The peak of hospitalizations for rotavirus gastroenteritis in the pre-vaccination period during the late winter and early spring coincides with the peak of infection with other human respiratory viruses (e.g., human respiratory syncytial virus, influenza, and adenovirus) among children under five, representing a possible cause of overloading pediatric wards and thus increasing the possibility of the spread of nosocomial infections, including RV [22]. According to a review carried out by STIKO, rotavirus vaccination can prevent even hospital-acquired rotavirus gastroenteritis [34].

RV presents high transmissibility due to the large number of viral particles eliminated in acute phases, the low infective dose (<100 viral particles), its long environmental persistence, and its relatively high resistance to disinfectants [33]. In particular, preterm infants who are admitted to NICUs from birth are more susceptible to RV nosocomial infections than term infants, which suggests that they should receive RV vaccines during admission and during periodic post-discharge follow-ups (NICU setting), according to the official immunization schedule [26,35].

The problem of transmission of vaccine virus to other preterm infants hospitalized in NICUs primarily involves the first vaccine dose, because it is unlikely that these infants remain in the hospital for more than 16 weeks, which is the last period for the second dose. The risk of transmission was previously evaluated as the shedding of RV in stool after vaccination with RV1 and RV5 in FT infants, and the results were dependent on the methods used to detect the virus [8,18,19]. In our study, the vaccine was administered on the ward in a minority of cases, so the risk of viral strain infection is minimal compared with that of vaccination.

Monk et al. conducted a retrospective study comparing gastrointestinal symptoms during the 15 days following vaccine administration in 96 vaccinated and 801 unvaccinated preterm infants from 2008 to 2010, examining the safety of RV5 in the NICU in age-eligible hospitalized preterm infants who could tolerate at least some enteral feedings. None of the preterm infants had symptoms that were attributed to RV5, although several infants in both groups had gastrointestinal manifestations. This result led the authors to conclude that the transmission of symptomatic infection to neighboring unvaccinated PT infants did not occur [36].

In our study, the overall rate of adverse events (AE) turned out to be twice as high as the data collected from the Italian pharmacovigilance surveillance system [37]. This evidence could be explained by the fact that the Italian pharmacovigilance surveillance system is typically passive compared to the active pharmacovigilance that was carried out in the present study. Moreover, the population enrolled in the Sicilian NICUs was limited and composed of particularly frail subjects.

RV vaccination in preterm infants, even during hospitalization, has been strongly recommended by scientific committees of different countries, such as the USA, Germany, Spain, and Sweden [38].

The American Committee of Immunization Practice (ACIP) considers the benefits of rotavirus vaccination of preterm infants (those born at <37 weeks gestation) to outweigh the risks of adverse events. ACIP supports vaccination of preterm infants according to the same schedule and precautions as full-term infants when the infant is clinically stable and the vaccine is administered at the time of discharge from the neonatal intensive care unit (NICU) or nursery, or after discharge from the NICU or nursery. However, in usual circumstances, the risk from shedding outweighs the benefit of vaccinating the infant who is age-eligible for vaccine but who will remain in the NICU or nursery after vaccination. [39].

In Germany, the STIKO Committee stated that the benefits of RV vaccination in neonatal intensive care units (NICU), providing protection against nosocomial RV infection, significantly outweigh the low risk of RV gastroenteritis in other hospitalized patients through nosocomial vaccine virus transmission in a joint statement with the German Academy for Pediatrics and Adolescent Medicine (DAKJ) and the German Society for Neonatology and Pediatric Intensive Care Medicine (GNPI) on RV vaccination of preterm infants and neonates during hospitalization [40].

In Spain, the Pediatric Spanish Association in 2022 recommended that preterm infants should be vaccinated without delay even if they are hospitalized, between 6 and 12 weeks post-birth [41].

Lastly, the Swedish Public Health Agency very recently recommended that preterm infants as well as children who are admitted for other reasons in the neonatal ward be vaccinated with the first dose against rotavirus infection when hospitalized and when age-eligible [42].

Our work presents some limitations: adverse events were reported by the parents of enrolled infants, and there was no control group. It was not possible to assess vaccine transmission within the NICU setting, primarily because fecal samples were not analyzed. However, data from the literature demonstrate the absence of transmission of vaccine rotavirus strains between vaccinated and unvaccinated infants in close proximity within a NICU, with no vaccine virus genomes detected in any of the stool samples collected from unvaccinated infants, unlike in vaccinated infants [43].

## 5. Conclusions

Data obtained from this study confirm the safety of the monovalent rotavirus vaccine even in preterm infants with gestational age ≥ 28 weeks, presenting an opportunity to improve the vaccination offer both in Sicily and in Italy by protecting the most fragile infants, who are more at risk of contracting severe rotavirus gastroenteritis and nosocomial RV infection.

Moreover, our data demonstrate the need for involving neonatologists in supporting rotavirus vaccination in Italian NICUs.

## Figures and Tables

**Table 1 vaccines-11-00718-t001:** Characteristics of preterm newborns enrolled at first rotavirus vaccination dose (n = 449).

	n (%)
**Sex, n (%)**	
-Male	223 (49.7)
-Female	226 (50.3)
**Average weight at first vaccination dose, grams ± SD**	3388.5 ± 903.4
**Feeding, n (%)**	
-Breastfeeding	35 (8.3)
-Artificial feeding	174 (41.3)
-Mixed feeding	212 (50.4)
**Comorbidities at birth, n (%)**	
-Moderate preterm birth (32 to 37 weeks)	241 (53.7)
-Very preterm birth (<32 weeks)	105 (23.4)
-Twinning	47 (10.5)
-Respiratory distress syndrome (RDS)	36 (8)
-Hypoglycaemia	10 (2.2)
-Hypocalcaemia	5 (1.1)
-Small for gestational age (SGA)	5 (1.1)

**Table 2 vaccines-11-00718-t002:** Characteristics of preterm newborns vaccinated against RV in the six Sicilian NICUs (neonatal intensive care units) involved in this study (n = 449).

	Newborns Enrolled n (%)	Average Gestational Age (Weeks) ± SD	Mean Age (Days) at First Administration ± SD	First Doses Administered in Hospital Ward vs. *Outpatient Hospital* Setting (Post-Hospitalisation) n (%)	Second Doses Administered in *Outpatient Hospital Setting* vs. Territorial Vaccination Clinics n (%)
**Buccheri La Ferla** **hospital (PA)**	15 (3.3)	33.4 ± 3.5	65.8 ± 15.2	11 (73.3)	3 (100)
**University Hospital of Palermo**	4 (0.9)	30.4 ± 1.8	42.7 ± 0.95	4 (100)	4 (100)
**ARNAS Civico Hospital (PA)**	119 (26.5)	33.5 ± 2.4	51.6 ± 10.1	11 (9.2)	85 (100)
**University Hospital of Catania**	58 (12.9)	32.3 ± 2.7	64.9 ± 18.2	0 (0.0)	36 (100)
**ARNAS Garibaldi hospital (CT)**	241 (53.7)	32.9 ± 4.3	53.1 ± 10.5	10 (4.2)	99 (41.8)
**University hospital of Messina**	12 (2.7)	34.7 ± 1.9	70.6 ± 9.1	0 (0.0)	6 (100)
**Overall**	**449**	**33.1 ± 3.8**	**55 ± 12.9**	**36 (8.1)**	**233 (62.8)**

**Table 3 vaccines-11-00718-t003:** Expected and unexpected adverse events (EAEs and UAEs) that occurred 14 and 28 days after first and second administration of monovalent rotavirus vaccine.

	Fever (≥38.5 °C) n (%)	Abdominal Colic n (%)	Diarrhoea n (%)	Vomiting n (%)	Intestinal Invagination n (%)	Food Refusal n (%)	UAEs n (%)
**EAEs 14 days after first administration**	0	2 (0.4)	5 (1.1)	0	0	0	0
**EAEs 28 days after first administration**	0	0	2 (0.4)	0	0	0	0
**EAEs 14 days after second administration**	1 (0.2)	1 (0.2)	0	0	0	0	0
**EAEs 28 days after second administration**	0	0	0	0	0	0	0

## Data Availability

Data obtained in the present study are available upon request to the corresponding author.

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
