# Peer review of "Safety of Rotavirus Vaccination in Preterm Infants Admitted in Neonatal Intensive Care Units in Sicily, Italy: A Multicenter Observational Study"

_vaccines, 2023, doi:10.3390/vaccines11040718_

Round 1

Reviewer 1 Report

1.       Title: Suggest removing “a real-life experience” in the title. It does not add any additional information.

2.       Abstract:

a.       Provide name of the vaccine used.

b.       “…were monitored 14 and 28 days after each of the two scheduled vaccine doses.” Unclear what is meant by this. At 14 and 28 day time points or for period 1-14 days and 15-28 days.

c.       Provide start and end date of the study.

d.       “in the 15 days after the first dose” – previously referred to 14 days.

e.       “No expected adverse effects were observed in the 28 days after both doses of the vaccine.” This is not in line with the results.  

f.        What is meant by “absolute safety”?

3.       General: the paper would benefit from improvement in the English language. E.g. Words like “amply” demonstrated and “remarkable” severity are quite vague and not really necessary as they do not add to the text. Can just say demonstrated and severity. “In a study by Vesikari at al. it was demonstrated a reduction in hospitalizations and emergency department visits related to G1-G4 rotavirus gastroenteritis occurring 14 or more days after the third dose by 94.5%  (95%CI, 91.2 to 96.6 percent) [7].” – sentence should say: “A study by Vesikari et al demonstrated…”. Line 114: “Data on” not “data of” safety. There are many other instances with minor grammatical errors or incorrect wording leading to misinterpretation, so please check throughout the paper.

Introduction:

4.       Line 105: define UMV. Make sure all abbreviations are defined on first use.

5.       Reference 11 is a case-control study investigating intussusception. Were hospitalisation rates pre and post vaccine introduction also assessed in the study or is the reference incorrect? Check all references for correctness.

6.       Line 110-111: “Right after the vaccine implementation, a decrease in the number of hospital admissions for RVGE was observed in Sicily among children under five years of age (12).” Provide the time period for the decrease. “Right after the vaccine implementation” is not specific and could mean anything. Reference 12 does not seem to be correct.

Methods:

7.       State that this is a descriptive study.

8.       Line 118: unclear what you mean by “real life context”. By stating that it is observational, you are distinguishing it from a clinical trail. Please clarify your meaning or remove.

9.       Did the 47 clinics where deliveries occurred refer babies who needed admission to the NICUs? How many NICUs are there in Sicily? Are these also private or public? Are the selected NICUs private or public? What about number of deliveries for the other 2 NICUs (numbers are only given for 4 of them)?

10.   Line 141: “antibiotic therapy for ep. ongoing infection” – what is the ep. referring to?

11.   Line 145 – provide definitions for expected, unexpected and serious adverse events.

12.   All the adverse events need to be clearly defined. E.g. diarrhea: how many stools over what time period? Same for irritability, skin inflammation, etc. How will abdominal pain and flatulence be objectively assessed in a neonate? What about fever? Were adverse events classified as mild, moderate and severe? Were the diary cards completed by the parent each day? Were all adverse events included in the diary card? How will the parent know what urticaria and intestinal invagination is? Were parents educated on how to complete the diary card? Was a thermometer provided to the parent to measure the temperature?

13.   Line 164: hesitancy or refusal. Was information collected on those that refused vaccination? Was consent obtained for data collection?

14.   Statistical analysis – This statistical analysis does not fit with your results. You do not present any data on hesitancy or refusal and do not present data from before and after the intervention. This is a descriptive study.

Results

15.   Line 172: 449 (out of 485 potentially enrollable due to clinical conditions) – do you mean that 36 were excluded due to presence of an exclusion criterion and thus not enrolled? Please clarify. Define RR.

16.   Line 175: Table 1 is a description of characteristics in the newborns. You are not comparing anything, only describing.  So what is meant by “There were no significant differences in gender”? Just state that there were 49.7% male and 50.3% female. You can only look at significant differences if you are comparing two groups.

17.   Table 3: Categorise adverse events as occurring 1-14 days after vaccination and 15-28 days after vaccination. Report the first and second doses separately.

18.   Line 199-200: You are reporting time to resolution of events. Was resolution of adverse events recorded in the diary card? This needs to be stated in the methods section.

19.   Line 200-201: “The clinical conditions of the children who reported adverse events were superimposable to those of the rest of the enrolled subjects.” Not clear what you mean by superimposable. Do you mean comparable? Where are these data shown? Provide in the appendix.

20.   Line 204: “The overall rate of adverse events was 161.8 per 100.000 doses administered.” How was this calculated. Needs to be in methods. E.g. Is total number of adverse events occurring 28 days after either vaccination per number of doses administered? Please clarify.

21.   Table 3 and line 196: How was the proportion of children with adverse events calculated? 1 out of 449 is 0.2%; 3/449 = 0.67%. Unclear how the proportions you are reporting were calculated.

22.   The discussion will need to be amended based on clarification of the results e.g. proportion of children with adverse events.

Author Response

Point 1: Suggest removing “a real-life experience” in the title. It does not add any additional information.

Response 1: We suggest as an alternative “a multicenter observational study”

Point 2:

  1. Provide name of the vaccine used.
  2. “…were monitored 14 and 28 days after each of the two scheduled vaccine doses.” Unclear what is meant by this. At 14 and 28 day time points or for period 1-14 days and 15-28 days.
  3. Provide start and end date of the study.
  4. “in the 15 days after the first dose” – previously referred to 14 days.
  5. “No expected adverse effects were observed in the 28 days after both doses of the vaccine.” This is not in line with the results.  
  6. What is meant by “absolute safety”?

Response 2: Please provide your response for Point 2. (in red)

  1. Added name RV1
  2. Adverse events (expected, unexpected, serious) were monitored from vaccine ad-ministration up to 14 (first assessment) and 28 days (second assessment) after each of the two doses of vaccine.
  3. From April 2018 to December 2019.
  4. 14
  5. Overall, 1.9% EAEs were observed at 14 days and 0.4% at 28 days after the first/second dose administration.
  6. You are right, we have changed it.

Point 3: General: the paper would benefit from improvement in the English language. E.g. Words like “amply” demonstrated and “remarkable” severity are quite vague and not really necessary as they do not add to the text. Can just say demonstrated and severity. “In a study by Vesikari at al. it was demonstrated a reduction in hospitalizations and emergency department visits related to G1-G4 rotavirus gastroenteritis occurring 14 or more days after the third dose by 94.5%  (95%CI, 91.2 to 96.6 percent) [7].” – sentence should say: “A study by Vesikari et al demonstrated…”. Line 114: “Data on” not “data of” safety. There are many other instances with minor grammatical errors or incorrect wording leading to misinterpretation, so please check throughout the paper.

Response 3: Thank you for the excellent comments, we have done our best to correct possible transcription errors, we remain available for other suggestions.

Introduction

Point 4: Line 105: define UMV. Make sure all abbreviations are defined on first use.

Response 4: Right, we added the meaning of the abbreviation. UMV: Universal Mass Vaccination

Point 5: Reference 11 is a case-control study investigating intussusception. Were hospitalisation rates pre and post vaccine introduction also assessed in the study or is the reference incorrect? Check all references for correctness.

Response 5: We apologize for the typo and thank you for the pertinent remark. In both cases ((even the one highlighted in Point 6) we were referring to reference 22; however, all references were checked.

Point 6: Line 110-111: “Right after the vaccine implementation, a decrease in the number of hospital admissions for RVGE was observed in Sicily among children under five years of age (12).” Provide the time period for the decrease. “Right after the vaccine implementation” is not specific and could mean anything. Reference 12 does not seem to be correct.

Response 6: Right, we changed to: “Since the implementation of vaccination starting in 2013, a progressive decrease in the number of hospital admissions for RVGE among children under five years of age has been observed in Sicily”.

Methods

Point 7: State that this is a descriptive study.

Response 7: Done.

Point 8: Line 118: unclear what you mean by “real life context”. By stating that it is observational, you are distinguishing it from a clinical trial. Please clarify your meaning or remove.

Response 8: Since we welcomed the title change, we have removed it.

Point 9: Did the 47 clinics where deliveries occurred refer babies who needed admission to the NICUs? How many NICUs are there in Sicily? Are these also private or public? Are the selected NICUs private or public? What about number of deliveries for the other 2 NICUs (numbers are only given for 4 of them)?

Response 9: In Sicily, each minor local health unit (Agrigento, Caltanissetta, Enna, Ragusa, Siracusa e Trapani) has its own NICU. The six NICUs shown are the only public NICUs in the three Sicilian main LHUs (Catania, Messina and Palermo) and collect most of the regional deliveries. There are no private NICUs; in Sicily, in fact, newborns from private clinics always belong to the public NICUs. Added also data from University Hospitals of Palermo and Messina (612 and 1317 deliveries, respectively).

Point 10: Line 141: “antibiotic therapy for ep. ongoing infection” – what is the ep. referring to?

Response 10: Thank you for noticing, the wording "ep" stood for "episode," we inserted the extended form: "antibiotic therapy for ongoing infectious episode."

Point 11: Line 145 – provide definitions for expected, unexpected and serious adverse events.

Response 11: In accordance with Directive 2001/20/EC Art. 2, adverse events are defined:

-Expected Adverse Events: Expected reactions, listed in the data sheet.

-Serious unexpected adverse event: a serious adverse reaction whose nature, severity, or outcome is not consistent with the reference safety information;

-Serious Adverse Event: any harmful clinical event that, regardless of the dose, requires hospitalization or prolongs ongoing hospitalization, results in severe or prolonged disability or incapacity, results in a congenital anomaly or birth defect, is life-threatening or causes death;

Point 12: All the adverse events need to be clearly defined. E.g. diarrhea: how many stools over what time period? Same for irritability, skin inflammation, etc. How will abdominal pain and flatulence be objectively assessed in a neonate? What about fever? Were adverse events classified as mild, moderate and severe? Were the diary cards completed by the parent each day? Were all adverse events included in the diary card? How will the parent know what urticaria and intestinal invagination is? Were parents educated on how to complete the diary card? Was a thermometer provided to the parent to measure the temperature?

Response 12: Active surveillance was carried out by parents, filling out a daily diary card that was returned at the time of the following outpatient visit or reported by telephone. If the child was hospitalized, active surveillance would be carried out directly at the hospital by the health care personnel. Parents were also asked to complete the diary card if the adverse events required pediatric counseling.  The diary card, was divided into sections, within which detailed instructions on how to fill them out properly were included. All adverse events were to be indicated as absent, mild (does not require pediatrician consultation), moderate (requires pediatrician consultation) or severe (if requires hospitalization).

Parents were informed and provided with informational materials on all Adverse Events in the data sheet, with their most common symptoms for diarrhea (stools that were less formed and/or more watery than usual were produced suddenly, and/or with some daily repetition), Urticaria (Reddened (erythema), raised (pomphi) and itchy areas that may also affect mucous membranes (e.g., lips and eyelids)), and intestinal invagination (vomiting, emission of stool mixed with mucus and blood, abdominal pain, palpable mass in the abdomen, and drowsy state), in order to more easily recognize and alert their pediatrician.

The assessment on abdominal pain, irritability and colic of the newborn is affected by the subjectivity of the parents; it cannot be objective.

Parents were also provided with a thermometer, to objectify possible systemic temperature elevations on a daily basis.

Point 13: Line 164: hesitancy or refusal. Was information collected on those that refused vaccination? Was consent obtained for data collection?

Response 13: We have informed consents with refusal to participate in the study from the parents of the 36 children who were not enrolled; we did not obtain informed consent to use their data.

Point 14: Statistical analysis – This statistical analysis does not fit with your results. You do not present any data on hesitancy or refusal and do not present data from before and after the intervention. This is a descriptive study.

Response 14: Thank you for the report, we removed the sentence, as it had been inserted by mistake.

Results

Point 15: Line 172: 449 (out of 485 potentially enrollable due to clinical conditions) – do you mean that 36 were excluded due to presence of an exclusion criterion and thus not enrolled? Please clarify. Define RR.

Response 15: The parents of thirty-six infants did not give consent, so they could not be enrolled. The overall RR was 92.6%. We changed the sentence as follows: “A total of 449 (of 485 potentially enrollable but whose parents did not provide consent; RR 92.6%)”

Point 16: Line 175: Table 1 is a description of characteristics in the newborns. You are not comparing anything, only describing. So what is meant by “There were no significant differences in gender”? Just state that there were 49.7% male and 50.3% female. You can only look at significant differences if you are comparing two groups.

Response 16: Thank you, we have corrected the sentence.

Point 17: Table 3: Categorise adverse events as occurring 1-14 days after vaccination and 15-28 days after vaccination. Report the first and second doses separately.

Response 17: Done.

Point 18: Line 199-200: You are reporting time to resolution of events. Was resolution of adverse events recorded in the diary card? This needs to be stated in the methods section.

Response 18: Certainly, thank you again, we have added this information in the materials and methods.

Point 19: Line 200-201: “The clinical conditions of the children who reported adverse events were superimposable to those of the rest of the enrolled subjects.” Not clear what you mean by superimposable. Do you mean comparable? Where are these data shown? Provide in the appendix.

Response 19: The clinical conditions of the children who reported adverse events were, at the pediatric examination that occurred at the time of enrollment, comparable to those of the rest of the enrolled subjects. We do not have a tabular data, but only a descriptive data of the visits made by pediatricians to the infants, which is not suitable for reporting in a possible appendix.

Point 20: Line 204: “The overall rate of adverse events was 161.8 per 100.000 doses administered.” How was this calculated. Needs to be in methods. E.g. Is total number of adverse events occurring 28 days after either vaccination per number of doses administered? Please clarify.

Response 20: Thank you for the observation, it is the overall number of adverse events occurring within 28 days after any dose, per number of doses administered. We have included it in the methods.

Point 21: Table 3 and line 196: How was the proportion of children with adverse events calculated? 1 out of 449 is 0.2%; 3/449 = 0.67%. Unclear how the proportions you are reporting were calculated.

Response 21: We are tremendously sorry for the error and we really thank you for pointing it out, we have corrected the percentages.

Point 22: The discussion will need to be amended based on clarification of the results e.g. proportion of children with adverse events.

Response 22: Done.

Reviewer 2 Report

The article under review is devoted to the actual problem of combating rotavirus infection in the department of newborns, especially when they are born prematurely. To combat this severe pathology, an appropriate vaccine is used. The authors set themselves the important task of assessing the safety of the use of this vaccine in the department for premature babies and successfully solved it by demonstrating a satisfactory safety of the vaccine. The work was well planned and executed. The results are adequate to the task. The conclusions are convincing and correspond to the task. There are no fundamental remarks to the work.

Author Response

Dear Reviewer, 
thank you for taking the time to read the article. We really appreciated the comments on the article. 

Sincerely

Authors 

Reviewer 3 Report

The article, while not appearing particularly original, provides interesting data about a topic of considerable interest. The number of infants who participated in the study is significant, and I appreciated the involvement of several units. However, the research has objective limitations, which the authors properly pointed out in the discussion.

The paper describes the experimentation at various stages, but for better understanding by readers, I think the information should be reorganized in the paragraphs more precisely.

Therefore, I report below some general and specific remarks, which I hope will improve the manuscript:

General proofreading of the draft for possible typos

I would include preterm infants in the keywords.

Line 62, I would insert the year of revision

The first part of the results paragraph (including Tables 1 and 2) describes and summarizes the observations and their characteristics, so I think it should be moved to the Materials and methods paragraph.

The results should be easily and quickly searchable

Line 214, I would insert the year of the study

Line 236, I would insert the year of the review.

With the hope that my comments will be helpful.

Regards,

the reviewer

Author Response

Dear Reviewer, 
thank you for taking time to read the article. We greatly appreciated the suggestions and have taken them into account in revising the article. 

Sincerely

Authors